# Characterization of Inducible HSP70 Genes in an Antarctic Yeast, *Glaciozyma antarctica* PI12, in Response to Thermal Stress

**DOI:** 10.3390/microorganisms9102069

**Published:** 2021-09-30

**Authors:** Nur Athirah Yusof, Jennifer Charles, Wan Nur Shuhaida Wan Mahadi, Abdul Munir Abdul Murad, Nor Muhammad Mahadi

**Affiliations:** 1Biotechnology Research Institute, Universiti Malaysia Sabah, Kota Kinabalu 88400, Sabah, Malaysia; jennifercharleslabo@gmail.com (J.C.); wanshuhaida94@gmail.com (W.N.S.W.M.); 2Faculty of Science and Technology, School of Biosciences and Biotechnology, Universiti Kebangsaan Malaysia, Bangi 43600, Selangor, Malaysia; munir@ukm.edu.my; 3Malaysia Genome Institute, Jalan Bangi, Kajang 43000, Selangor, Malaysia; nor.mahadi@gmail.com

**Keywords:** HSP70 chaperone, Antarctica, heat stress, heat shock proteins, cell adaptation, *Glaciozyma antarctica*

## Abstract

The induction of highly conserved heat shock protein 70 (HSP70) is often related to a cellular response due to harmful stress or adverse life conditions. In this study, we determined the expression of *Hsp70* genes in the Antarctic yeast, *Glaciozyma antarctica*, under different several thermal treatments for several exposure periods. The main aims of the present study were (1) to determine if stress-induced *Hsp70* could be used to monitor the exposure of the yeast species *G. antarctica* to various types of thermal stress; (2) to analyze the structures of the *G. antarctica* HSP70 proteins using comparative modeling; and (3) to evaluate the relationship between the function and structure of HSP70 in *G. antarctica*. In this study, we managed to amplify and clone 2 *Hsp70* genes from *G. antarctica* named *GaHsp70-1* and *GaHsp70-2*. The cells of *G. antarctica* expressed significantly inducible *Hsp70* genes after the heat and cold shock treatments. Interestingly, *GaHsp70-1* showed 2–6-fold higher expression than *GaHsp70-2* after the heat and cold exposure. ATP hydrolysis analysis on both *G. antarctica* HSP70s proved that these psychrophilic chaperones can perform activities in a wide range of temperatures, such as at 37, 25, 15, and 4 °C. The 3D structures of both HSP70s revealed several interesting findings, such as the substitution of a β-sheet to loop in the N-terminal ATPase binding domain and some modest residue substitutions, which gave the proteins the flexibility to function at low temperatures and retain their functional activity at ambient temperatures. In conclusion, both analyzed HSP70s played important roles in the physiological adaptation of *G. antarctica*.

## 1. Introduction

HSP70 proteins in marine organisms have gained much research attention. They serve as potential biomarkers for environmental stress because of their high sensitivity and variability in expression when cells are exposed to different stimuli [1,2]. They also play important roles in cellular protein folding processes, change the structures of newly synthesized proteins, and repair unsuccessful protein conformations [3]. In bacteria, DnaK is a bacterial representative of the HSP70 chaperone family, which plays a key role in protein folding during non-stress growth conditions and becomes more important during stress conditions [4]. To elucidate the characteristics of the HSP70, numerous studies have been conducted using HSP70 as a biomarker to signal the possible environmental changes expressed in the organisms through the expression of the genes and the subsequent activation of the protein [5,6,7,8,9,10,11]. Furthermore, HSP70s are the best indicator for environmental stress, such as heat, because they are activated in all cellular reactions, reside in all subcellular compartments, are abundant and highly sensitive, are extremely important for cellular repair machinery, and are involved in the homeostasis of all living organisms [12,13,14,15].

From a molecular point of view, the mechanisms of action of HSP70 are generally based on the N-terminal ATPase domain and the C-terminal binding domain, which is subdivided into a β-sandwich subdomain and a C-terminal α-helical subdomain. The sequence alignment of the HSP70 proteins reveals differences that may reflect adaptations related to resistance and resilience to environmental stress [16]. A shift in the ATPase cycle can be observed, with it changing between the ATP state, which has a low affinity and a rapid energetic cost for a substrate, and ADP state, which has a high affinity and a low energetic cost for the substrate [3]. Although the importance of HSP70 is undeniable, research on the structure and function of psychrophilic HSP70 is still in its infancy. Hence, it is important to unveil the relationship between the structure and function of HSP70 in order to understand the response and level of resistance towards thermal stress in living organisms in Antarctica.

*Glaciozyma antarctica* PI12 is an obligate psychrophilic yeast isolated from the Antarctic marine water near Casey Station (S66° 21′ 25″; E110° 37′ 09″) in the Southern Ocean of Antarctica. It can grow in a range of temperatures from −12 °C to 20 °C, with its optimal growth temperature being 12 °C. [17]. The genome data have revealed that *G. antarctica* PI12 has a total of 7857 protein-coding genes [18]. Previously, Boo et al. (2013) measured the expression levels of genes in *G. antarctica* PI12 that was exposed to cold and heat shocks using quantitative real-time PCR (qPCR) [17]. Several related studies on *G. antarctica* PI12 have provided some insight into the strategies used by this yeast to adapt to thermal stress [19,20,21,22,23]. Most stress-related gene studies have been carried out at the gene level, and our understanding of the molecular architecture of the proteins is still limited [5,6,7,9,13,17,24,25,26,27,28,29]. In the previous study by Yusof et al. (2019), a total of five *Hsp70s* were found to have significant expression patterns in response to heat stress [23]. *G. antarctica Hsp70* genes were identified to contain transcript-1 and transcript-2, which were found in the ribosome and mitochondria, respectively [23]. Meanwhile, in this study, the full lengths of 2 out of 5 *Hsp70* genes were amplified and characterized. Transcript-1 and transcript-2 were characterized and named *GaHsp70-1* and *GaHsp70-2*, respectively. Therefore, the findings from this study will provide the community with a better understanding of the adaptation mechanisms of *G. antarctica* PI12 by determining the relationship between the structure and function of the HSP70 stress proteins.

In the present study, *G. antarctica* PI12 was chosen for several reasons. Firstly, the availability of its genome allows the thorough investigation of its heat shock proteins. Secondly, it can be readily maintained under laboratory conditions for in vitro and in vivo experiments, as many studies have previously proven [18,20,21,22]. HSP70′s protein-coding genes were the main targets, since the *Hsp70* genes appeared to be significantly expressed when exposed to a sublethal temperature of 22 °C [17]. Moreover, *G. antarctica* PI12 has a high similarity with *Saccharomyces cerevisiae*, with more than 60% similarity; hence, it may serve as a good model organism with which to study the thermal adaptation of the eukaryotic polar organism. Since the genome of *G. antarctica* PI12 and the optimal cultivation conditions are known, *G. antarctica* PI12 may serve as an excellent model organism for the polar environment. Therefore, using gene expression analysis with real-time polymerase chain reaction (PCR), the aims of the present study were (1) to determine if stress-induced *Hsp70* could be used to monitor the exposure of the yeast species *G. antarctica* to various types of thermal stress, (2) to analyze the structures of the *G. antarctica* HSP70 proteins using comparative modelling, and (3) to evaluate the relationship between the function and structure of HSP70 in *G. antarctica*.

## 2. Results

### 2.1. G. antarctica HSP70 Sequence Analysis

Extracted RNA was analyzed to determine its integrity and the presence of 18 S and 28 S using gel electrophoresis (Appendix A). *GaHsp70-1* and *GaHsp70-2* were amplified at the target sizes of 1884 bp and 2019 bp, respectively, followed by cloning and verification using sequencing (Appendix A). The *G. antarctica* HSP70 sequence analysis revealed that these proteins are related to HSP70 proteins from other organisms. GaHSP70-1 was found to be related to HSP70s from *Rhodotorula graminis* WP1 (acc. no. XP_018267745.1), *Rhodotorula diobovata* (acc. no. TNY21258.1), *Rhodotorula* sp. JG-1b (acc. no. KWU41802.1), and *Rhodotorula mucilaginosa* (acc. no. KAG0657616.1), with a percentage identity between 81% and 91%. In addition, GaHSP70-2 was found to be related to HSP70s and DnaK from *Leucosporidium creatinivorum* (acc. no. ORY92759.1), HSP70-like protein *Microbotryum lychnidis-dioicae* p1A1 Lamole (acc. no. KDE08371.1), *Rhodotorula toruloides* NP11 (acc. no. XP_016269663.1), and *Puccinia sorghi* (acc. no. KNZ63140.1), with a percentage identity between 73% and 89% (Appendix A). The sequence alignment with model organisms showed that both sequences contained a C-terminal subdomain, substrate-binding domain, and conserved ATPase N-terminal binding domain (Appendix A). Based on the GO analysis, both HSP70s were projected to have a similar biological process for protein folding (GO:0006457) and molecular functions for ATP binding (GO: 0005525) and ATPase (GO: 0016887). Phylogenetic tree analysis showed that GaHSP70-1 and GaHSP70-2 have a high similarity at the protein level with other HSP70s from other fungi (Figure 1).

### 2.2. G. antarctica Hsp70 mRNA Expression Levels at Different Temperatures

The real-time PCR primers were designed to specifically amplify *G. antarctica Hsp70* at the level of 100 to 200 base pairs. The efficiency of the primers used in this study had values of between 95% and 110%, with R^2^ values ranging from 0.97 to 1.00 (Table 1). The purity of the RNA was 1.9 to 2.0. The melting curve analysis showed a single amplification of the *18 S*, *GaHsp70-1*, and *GaHsp70-2* genes. The PCR efficiency of the primers used to amplify the *18 S*, *GaHsp70-1,* and *GaHsp70-2* genes was acceptable, as the slopes were within the range of −3.0 to −3.3. Thus, the template purity and PCR efficiency fulfilled the standard characteristics [30]. The *GaHsp70-1* and *GaHsp70-2* messenger RNA (mRNA) expressions in *G. antarctica* were measured at different temperatures to determine the induction of *G. antarctica Hsp70* genes (Figure 2). *GaHsp70-1* mRNA expression strongly increased in cells exposed to temperatures far from the optimal growth of 12 °C, with temperatures of −20, 0, 20, and 30 °C associated with 5.26-, 6.33-, 5.43-, and 4-fold increases, respectively. Meanwhile, the *GaHsp70-2* mRNA expression showed increments of 2.82-, 1.3-, 2.43-, and 1.37-fold, lower than the *GaHsp70-1* mRNA expression at −20, 0, 20, and 30 °C. The increase in expression of both *G. antarctica Hsp70* mRNAs at all exposure temperatures indicates the importance of *Hsp70* in cell stress management in conditions of thermal stress.

### 2.3. ATPase Assay Analysis

In order to determine the ability of GaHSP70-1 and GaHSP70-2 to hydrolyze ATP at temperatures lower and higher than room temperature, ATPase activity assays were conducted to measure the rate of ATP hydrolysis at 37, 25, 15, and 4 °C (Figure 3). The measurement of the ATPase activity using a malachite green colorimetric assay is commonly carried out at room temperature, as this is the optimum temperature for ATP hydrolysis [24]. The by-product of the ATP hydrolysis, which was the phosphate ions in this case, was measured. One unit is the amount of ATPase that catalyzes the production of 1 mole of free phosphate per minute under the assay conditions. In this study, the ATPase activities in GaHSP70-2 were slightly higher compared to those in GaHSP70-1 at all temperatures tested, except 15 °C. Both GaHSP70-1 and GaHSP70-2 showed approximately the same rate of ATP hydrolysis at 15 °C, which was 0.516 µmole/min/µL. At temperatures lower than 25 °C, the ATP hydrolysis activity decreased almost by two-fold compared to the activity at 25 °C. This finding showed that GaHSP70-1 and GaHSP70-2 could function at a lower activity rate at cold temperatures below 25 °C.

### 2.4. 3D Structure Analysis

GaHSP70-1 and GaHSP70-2 had a 60% similarity with each other. GaHSP70-1 had the highest similarity with human HSP70 (PDB: 5FPN), with a 73.44% similarity with an E-value of 0. Meanwhile, GaHSP70-2 had the highest similarity of 60.27% with *E. coli* DnaK (PDB: 5NRO), with an E-value of 0. The tertiary structures of GaHSP70-1 and GaHSP70-2 were modelled with SWISS-MODEL using the structures of 5FPN and 5NRO as templates, respectively [31,32,33,34,35,36,37,38,39]. Both models were evaluated using PROCHECK, Verify3D, and ANOLEA. PROCHECK analysis showed that the build models had 100% amino acids in favored and allowed regions. Furthermore, model verification using Verify3D showed that the constructed models obtained a positive score of more than 80%. Analysis using ANOLEA showed acceptable energy calculations at the atomic level in the protein model structure. The superimposed GaHSP70-1 model, and its template produced a very significant RMSD value of 0.071 Å. A comparative analysis of the GaHSP70-1 model and 5FPN showed that the constructed model contained a substitution of a β-sheet to a loop located at the N-terminal ATPase binding domain (Figure 4). In addition, the superimposed GaHSP70-2 model, and its template gave a significant RMSD value of 0.118 Å. The sequence analysis also revealed several types of residue substitution in GaHSP70-2: (1) substitution to alanine and (2) the substitution of charged and polar amino acids to uncharged and hydrophobic amino acids. The residue substitutions to alanine were identified at the ATP-binding domain at positions 106, 118, 228, 299, 321, 364, 388, 416, and 462. There were two substitutions to alanine at the substrate-binding domain, which were at 572 and 577. Besides these, several charged and polar amino acids in the ATP-binding domain were replaced with uncharged and hydrophobic amino acids. For example, arginine at position 61 was substituted to isoleucine, aspartic acid at position 90 was substituted to leucine, serine at position 154 was substituted to glycine, aspartic acid at position 257 was substituted to asparagine, and histidine at position 272 was substituted to tyrosine. There were two residue substitutions in the substrate-binding domain; negatively charged residues were substituted for uncharged ones at position 475 (glutamic acid substituted for glutamine) and a bulky polar side chain was substituted for a hydrophobic side chain at position 493 (serine substituted for valine). Three-dimensional structure analysis revealed that these residue substitutions increased the distance of interaction between the two residues. For example, the substitution of valine for alanine at position 364 in GaHSP70-2 weakened the aromatic interaction that occurred at 3.441 Å (Figure 5).

## 3. Discussion

Previous studies have proven the vital role of HSP70 in the stress resistance of organisms, which is mostly related to temperature stress [41]. This study is the first description of the gene expression of HSP70 chaperones of the Antarctic obligate psychrophilic yeast, *G. antarctica* PI12, during conditions of acute thermal stress and cold shock. To date, several investigations on Antarctic organisms have evaluated the thermal stress response related to the expression of *Hsp70* at different temperatures [26,27,28,29,42,43,44,45]. However, in Antarctic yeast, the expression of type HSP70 chaperones has not yet been evaluated. Hence, this study in Antarctic yeast is crucial in order to understand how these organisms face the current and future changes in the Antarctic seawater temperature as well as to possibly measure the effects of different climate change scenarios.

In this study, we cloned full-length cDNA of 1815 bp and 2019 bp encoding HSP70 in *G. antarctica* PI12. The lengths of the *GaHsp70-1* and *GaHsp70-2* ORFs were 627 and 672 amino acids, respectively. Multiple sequence alignment revealed that HSP70 in *G. antarctica* contained conserved regions found in *S. cerevisiae*, *A. thaliana*, *D. melanogaster*, and *Homo sapiens*. Structurally, the GaHSP70-1 and GaHSP70-2 contained a highly conserved ATPase binding domain for ATP hydrolysis, a substrate-binding domain, and a varied C-terminal region that functioned as a lid. The phylogenetic analysis showed that both *G. antarctic* HSP70s have a high homology with HSP70s from other organisms, reflecting its long independent evolutionary history. GaHSP70-1 and GaHSP70-2 are possibly paralogs present in the *G. antarctica* genome that arise by duplication, since they have a high sequence similarity and common ortholog proteins with HSP70 from other organisms. Both HSP70 proteins are diversified into heat-inducible paralogs; their functions in stress response may vary significantly from a low effect to a rapid effect and strong protein folding activities [46,47,48,49].

Our study on *Hsp70* expression patterns showed that the expression levels of both *GaHsp70-1* and *GaHsp70-2* increased as the temperature increased, and all were higher than those of the control groups. Similar results were reported by previous studies that showed the upregulation of *Hsp70* mRNA expression levels when cells were exposed to higher temperatures [1,28,44,50]. Interestingly, the expression level of *GaHsp70-1* was higher than that of *GaHsp70-2* by up to 6-fold. *GaHsp70-1* showed a 2-fold higher expression compared to *GaHsp70-2* when exposed to 20 °C and a 4-fold higher expression when exposed to 30 °C. At −20 °C, the expression level of *GaHsp70-1* increased by 5.26-fold while that of *GaHsp70-2* increased by 2.82-folds, which was about 2-fold higher than that of *GaHsp70-1*. At 0 °C, the expression of *GaHsp70-1* was 6-fold higher than that of *GaHsp70-2*. The importance of HSP70 in bacteria to eukarya has been demonstrated in numerous studies [50,51,52,53,54,55]. One study showed the abundance of HSP70 in the psychrophilic bacterium *Shewanella frigidimarina* when cells were exposed to heat stress at 28 °C [56]. Another study on the Antarctic haloarchaea showed that *Halohasta litchfieldiae* and *Halorubrum lacusprofundi* had higher expression levels of *Hsp70* at high temperatures in response to heat stress [55]. In addition, studies in *D. melanogaster*, *Caenorhabditis elegans*, rodents, insects, and humans have also indicated that the higher the expression level of HSP70, the better the thermal stress tolerance of cells [50,51,52,53,54]. Rout et al. (2016) indicated that the gene expression pattern of *Hsp70* in different goat organs exhibited a higher expression than that of the control during the peak heat stress period [57]. These findings confirmed the importance of HSP70 in protecting cells against thermal and cold stress. Hence, HSP70 serves as a potential biomarker for determining the expression of mRNA inside *G. antarctica* cells upon exposure to thermal stress.

At the protein level, we decided to determine whether GaHSP70-1 and GaHSP70-2 were able to function at temperatures lower and higher than the *G. antarctica* optimum growth temperature, which is 12 °C. Since this is the first study, we know of to be carried out on the Antarctic yeast HSP70, we decided to test the *G. antarctica* HSP70 function at temperatures ranging between 4 °C and 37 °C. The results showed that both GaHSP70-1 and GaHSP70-2 were able to function at the tested temperatures. Both HSP70 proteins had similar assay values at 15 °C. The analysis of the ATP assay at 37, 25, and 4 °C showed that GaHSP70-2 was able to perform better in ATP hydrolysis than GaHSP70-1. Therefore, this finding suggested that GaHSP70-2 plays a more important role than GaHSP70-1 in protein folding and the prevention of aggregation processes in cells, especially during thermal stress. At the cellular level, heat stress causes the denaturation of polypeptides, while cold stress causes degradation. Cells activate HSP70 to prevent and minimize the accumulation of denatured toxic and abnormal proteins [1,14,58]. This reflects the importance of HSP70 in response to thermal and cold stress in *G. antarctica*.

The determination of the 3D protein structure of HSP70 from *G. antarctica* PI12 was crucial to elucidate the relationship between HSP70′s structure and function, thus helping us to understand the mechanisms of thermal adaptation in cold-adapted proteins [59]. The protein structure of GaHSP70-1 has a high similarity with that of human HSP70, with a very significant RMSD of 0.071 Å. Superimposing the two protein structures with an RMSD of below 2 Å shows their high similarity, as the proteins share a similar architecture. The analysis of the GaHSP70-1 structure showed a substitution of the β-sheet with a loop at the ATP-binding domain (Figure 4). To date, this is the first study to report the replacement of a β-sheet with a loop at the ATP-binding domain of a cold-adapted HSP70. Structural flexibility and dynamics are strongly related to the presence of loops, which increased the degree of structural flexibility and contributed to the increase in enzyme activity [60,61]. In cold adaptation studies, loops in protein structures have been associated with better flexibility as a central role in adaptation to cold environments [62]. The extension of a loop in the protein structure is also associated with structural adaptation in cold-adapted proteins [63]. Hence, the presence of a loop at the ATP-binding domain may promote flexibility and thus allow ATP hydrolysis at low temperatures. Moreover, our analysis on HSP70-2 showed that the protein was structurally more similar to DnaK in *E. coli*, with a 60.27% identity. The superimposing of the GaHSP70-2 model with a DnaK structure achieved an RMSD of 0.118 Å, which indicated that both structures were aligned closely, as they shared a high similarity in terms of their protein structure. The protein architecture of GaHSP70-2 revealed several residue substitutions with alanine as well as less polar and charged residues at the ATP-binding domain. Cold-adapted enzymes generally exhibit an increased glycine content and reduced arginine, proline, and acidic amino acid contents [64]. Structural studies on cold adaptation have also revealed that a modest substitution with alanine increased the structural ability of a protein to function in the cold while maintaining its rigidity and stability [65]. The analysis of the protein structure at the atomic level showed that several substitutions could hinder the strong force and contribute to protein rigidity. For example, the substitution of valine for alanine at position 364 in GaHSP70-2 hindered the strong aromatic interaction with phenylalanine at 3.581 Å. Aromatic interactions provide stability to protein structures and the weakening of aromatic interactions improves structural flexibility in cold-adapted enzymes [66]. Hence, we propose that the substitutions of loop and alanine at the HSP70 ATP-binding domain may possibly contribute to the structural flexibility, which is important for cold adaptation. The substitution of polar, charged residues for nonpolar, uncharged, and hydrophobic residues is another cold adaptation characteristic of *G. antarctica* HSP70 that has been reported in several studies on cold-adapted enzymes [67].

The suite of amino acid substitutions observed in the *G. antarctica* HSP70s is consistent with the evolution of polypeptide flexibility to facilitate efficient folding activity for cold adaptation. Another question that has not been answered yet is how cold-adapted proteins respond to heat stress, especially during climate change. It could be possible that the residue substitutions observed in the HSP70 protein structures have evolved and undergo positive selection, allowing them to not only function at low temperatures but also to retain their functions during thermal stress, thus protecting the host organisms from cold denaturation and heat aggregation. In our study, the HSP70 genes in *G. antarctica* PI12 were significantly expressed at high and low temperatures and functioned at a wide range of temperatures from 4 °C to 37 °C. Our findings showed that *G. antarctica* HSP70s have evolved to function at low temperatures and retained their functions during thermal stress. Hence, these HSP70s may serve as potential biomarkers for detecting the effects of global climate change in the Antarctic Ocean.

## 4. Materials and Methods

### 4.1. Culturing G. antarctica and Exposure to Different Heat and Cold Shock Temperatures

The isolated *G. antarctica* PI12 culture was obtained courtesy of the School of Biological Sciences, Universiti Sains Malaysia. The cells were cultured in yeast peptone dextrose broth (10% (*w*/*v*) yeast extract (Sigma-Aldrich, St. Louis, MO, USA), 20% (*w*/*v*) peptone (Sigma-Aldrich, USA), and 20% (*w*/*v*) dextrose (Sigma-Aldrich, USA)) in shake flasks at 200 rpm and 12 °C until the OD_600_ reached approximately 0.6–0.8. Subsequently, the cultures were then exposed to different temperatures of −20, 0, 12, 20, and 30 °C. The cells were harvested at 10,000 rpm for 10 min after 6 h of exposure to each temperature and the pellet was stored at −80 °C until further usage.

### 4.2. Mining Hsp70 Genes from G. antarctica Genome

The nucleotide sequences of the *Hsp70* genes from *G. antarctica* were designated *GaHsp70-1* and *GaHsp70-2*. The taxonomy identity for *G. antarctica* PI12 in NCBI is NCBI:txid1332765. The *Hsp70* sequences were obtained from the genome data of *G. antarctica* (GenBank assembly accession GCA_002917775.1); a total of 7857 protein-coding genes have been identified in this Antarctic yeast [18]. The degree of homology of these *Hsp70* sequences was confirmed by similarity search using the Blastx program NCBI (https://blast.ncbi.nlm.nih.gov/Blast.cgi; (accessed on 3 August 2021)). The HSP70 amino acid sequences were aligned using a multiple sequence alignment by the ClustalW program (https://www.genome.jp/tools-bin/clustalw; (accessed on 3 August 2021)) to analyze regions with similarities with other organisms [68]. Subsequently, the primers were designed using the software Primer 3.0.

### 4.3. RNA Extraction

The total RNA of *G. antarctica* PII12 after exposure to different temperatures was extracted using the GeneJET RNA Purification kit (Thermo Scientific, Waltham, MA, USA) following the manufacturer’s protocol. Before the extraction yeast lysate was prepared, the yeast cell pellet was resuspended in Yeast Lysis buffer (1 M sorbitol (Sigma-Aldrich, USA), 0.1 M EDTA (Sigma-Aldrich, USA), pH 7.4, 0.1% -β-mercaptoethanol (Sigma-Aldrich, USA), and zymolyase 20T (Sigma-Aldrich, USA)) and incubated at 30 °C for 30 min. Then, the yeast sample was lysed with Lysis Buffer (Thermo Scientific, USA) supplemented with 14.3 M β-mercaptoethanol and Proteinase K (Thermo Scientific, USA) diluted with TE buffer (10 mM Tris HCl, pH 8.0, 1 mM EDTA) and incubated at 22 °C for 10 min. Following extraction, the concentration and purity of the total RNA were measured using a Nanodrop spectrophotometer (Thermo Scientific, USA) at 260 nm. The RNA quality was evaluated on 1% (*w*/*v*) agarose (Nacalai, Kyoto, Japan) gel electrophoresis at 100 V for 20 min and kept at −80 °C until further use.

### 4.4. Synthesis of cDNA

Before the cDNA synthesis, the extracted RNA was treated with DNase I (Thermo Scientific, USA) to remove traces of genomic DNA contamination from the RNA sample. The pretreatment protocol was conducted based on the manufacturer’s protocol. Both the gDNA removal and first-strand cDNA synthesis were performed using the ReverTra Ace™ qPCR RT Master Mix with a gDNA Remover (TOYOBO, Osaka, Japan). A total of 10 μL of reaction solution containing 8 μL of reaction solution from the DNase I treatment (2 μL mixture of 1 in 50 volume of gDNA remover to 4× DN Master Mix, 0.5 μg of total RNA template and nuclease-free water) with 2 μL of 5× RT Master Mix II (master mix reagent contains oligo dT and random primers). The reaction was carried out in the PCR thermal cycler (Bio-Rad, Hercules, CA, USA) at 37 °C for 15 min, followed by 50 °C for 5 min, and was terminated at 98 °C for 5 min. Then, the first-strand cDNA was used as the template for full-length cDNA amplification. Specific primers were used to amplify the cDNA targeting *GaHsp70-1* and *GaHsp70-2* (Table 2). The PCR mixture contains 200 ng of cDNA template, 20 μM forward and reverse primers, and 1× MyTaq Red Mix (Bioline, London, UK). PCRs were performed as follows: Initial denaturation at 95 °C for 1 min, followed by 35 cycles of denaturation at 95 °C for 15 s, annealing at 60 °C for 15 s, and elongation at 72 °C for 10 s. The reaction was terminated at 72 °C for 10 min. Subsequently, the PCR product was analyzed using agarose gel electrophoresis (1% *w*/*v*) at 100 V for 40 min.

### 4.5. mRNA Expression Analysis by Real-Time PCR

Quantitative PCR analysis was carried out to determine whether there were acute changes in gene expression; sampling occurred after 6 h exposure during the sublethal and acute thermal stress. The RT-qPCR reactions were carried using a real-time thermal cycler, CFX96 (Bio-Rad, USA). The assays were run using SensiFAST SYBR^®^ No-ROX Kit (Bioline, UK). Each real-time PCR mixture contained 10 μL 2× SensiFAST SYBR^®^ No-ROX Mix, 0.8 μL of a 10 μM forward/reverse primer, and ~100 ng cDNA template (refer to 3.4.2 for cDNA preparation). This was lastly topped up with RNase-free water to obtain a final volume 20 μL. Specific primers were designed based on sequence data from the *G. antarctica* database. A total of three sets of primers, which included *GaHsp70-1* and *GaHsp70-2* and *18 S*, were used in the reactions (Table 2). The real-time cycler conditions were set as follows: reverse transcription step at 45 °C for 10 min for 1 cycle; an initial activation step at 95 °C for 2 min; 50 cycles of the denaturation step at 95 °C for 10 s; annealing at 60 °C for 10 s; and then exposure to 72 °C for 10 s. At the end of each run, the melting curves for the amplicons were determined by raising the temperature during fluorescence monitoring by 0.4 °C from 60 °C to 95 °C. The setpoint temperature was increased after cycle 2 by 0.5 °C. Subsequently, 2% agarose gel electrophoresis and melting curve analysis confirmed the PCR amplification. A standard curve was constructed using 10-fold serial dilutions (100, 10, 1, 0.1, and 0.01 ng) of RNA amplified with *18 S* reference gene and *GaHsp70* primers. The analysis was conducted in triplicate. A one-way analysis of variance (ANOVA) at a significance level of *p* < 0.05 was used to compare the expression levels at different temperatures. The *GaHsp70* expression profiles were normalized with the 18 S to compensate for any variation in the amount of starting material among the samples [18,20]. Melting curves analyses of the PCR reactions were carried out to analyze the specificity of each set of primers.

### 4.6. Cloning and Sequence Analysis

The cDNA amplification of *GaHsp70-1* and *GaHsp70-2* was conducted, and they were cloned into pGEMT-T easy vector (PROMEGA, Madison, WI, USA). Finally, the annealed product was transformed into an *E. coli* DH5α cloning host using the heat shock method. For protein expression, specific primers promoting the ligation-independent cloning of target genes into the pET30 Ek/LIC (Merck-Millipore, Germany) vector were used (Table 3). The direct ligation of amplified *GaHsp70-1* and *GaHsp70-2* with the pET30 Ek/LIC vector was carried out according to the manufacturer’s instructions. Moreover, the annealed products were transformed into an *E. coli* BL21(DE3) expression host. The positive transformants were identified using sequencing analysis. The isoelectric point was determined using the ProtParam tool [69]. The sequence domain was analyzed using InterPro Scan [70] and Pfam [71]. Sequence alignment was performed using ClustalW [72]. A phylogenetic tree analysis was carried out by the neighbor-joining method in MEGA X and the confidence values among species were calculated by bootstrapping repeated 1000 times [73,74]. The sequences of the *GaHsp70-1* and *GaHsp70-2* were deposited in the GenBank™ database under these accession numbers of AEG19530.1 and MZ313862, respectively.

### 4.7. Expression, Purification, and Detection of Recombinant G. antarctica Hsp70

Expression was assessed in LB containing 50 μg/mL of kanamycin (Nacalai, Japan). Induction was performed using 0.5 mM isopropyl β-D-thiogalactopyrosidase (IPTG) (Nacalai, Japan) after reaching an OD_600_ of ~0.8 and inducing for 16 h at 20 °C. The overexpression of the *G. antarctica* HSP70 was analyzed using 15% SDS-PAGE (Appendix A). The protein bands were visualized by Coomassie brilliant blue R250 staining (Nacalai, Japan). Purification was performed using prepacked Ni-NTA columns and AKTA Avant based on the manufacturer’s instructions (GE Healthcare, Marlborough, MA, USA). The purified GaHSP70-1 and GaHSP70-2 were analyzed for purity using SDS-PAGE (Appendix A).

### 4.8. Hsp70 ATPase Activity Analysis

The ATPase activity was evaluated by the measurement of colorimetric product resulting from the malachite green reagent and free ion [PO_4_]^3−^ measured at 620 nm. The ATPase assays were carried out using an ATPase/GTPase Activity Assay Kit (Sigma-Aldrich, USA). The assay reaction mixture was composed of 10 µg recombinant protein (for the ATPase activity measurements) incubated in a 30 uL reaction volume containing 20 µL 40 mM Tris, 80 mM NaCl, 8 mM MgAc2, 1 mM EDTA at pH 7.5, and 10 uL 4 mM ATP (Sigma-Aldrich, USA). To monitor the effect of temperature on the activity of GaHSP70-1 and GaHSP70-2, the ATPase reaction mixture was incubated at 37, 25, 15, and 4 °C for 1 h. After 1 h of incubation, the reaction was stopped by adding 200 µL of malachite green reagent and incubated for an additional 30 min at room temperature to generate the colorimetric product. The product mixtures were loaded onto a 96-well plate and the absorbance values of colorimetric products were read using a SpectraMax spectrophotometer (Molecular Devices, San Jose, CA, USA) at 620 nm. All the samples were run in triplicate. Phosphate standard values for colorimetric detection were prepared according to the manufacturer’s instructions.

### 4.9. Modeling GaHSP70 Tertiary Structures

The three-dimensional GaHSP70-1 and GaHSP70-2 were modelled for the *Escherichia coli* DnaK (PDB entry 5NRO) and human HSP70 (PDB entry 5FPN), respectively, using the SWISS-MODEL program [31,32,33,34,35,36,37,38,39]. The structure quality was evaluated using PROCHECK [75], Verify3D [76], and ANOLEA [77,78,79]. The superimposed model and template and comparative analyses were performed using CHIMERA USCF [40].

## 5. Conclusions

In the present study, we report the complete cDNA of two *Hsp70s* encoding the ORFs of *G. antarctica*, as well as their amino acid sequences and protein structure models. The questions addressed in this study were whether *G. antarctica* PI12 has a significant effect on the gene regulation of *Hsp70* during thermal and cold stress conditions and if the protein structures confer certain traits that allow thermal and cold adaptation ability in the Antarctic yeast *G. antarctica*. Our results provide the first molecular evidence that both *GaHsp70-1* and *GaHsp70-2* are significantly upregulated and inducible by thermal and cold stress in *G. antarctica* PI12. These findings are similar to responses shown in many Antarctic organisms, such as the Antarctic marine mollusc (*Laternula elliptica* and *Nacella concinna*), producing a rapid increase in the *Hsp70* response. The protein structure models showed residue substitution strategies that allowed the protein to function well at a wider range of temperatures, including low and high temperatures. Therefore, this study indicates that the residue substitutions observed in the *G. antarctica* HSP70 structure have evolved and undergo positive selection to not only function at low temperatures but also to retain their functions during thermal stress, thus protecting host organisms from cold denaturation and heat aggregation. We unveil the underlying transcriptional plasticity and functional divergence of HSP70 copies in *G. antarctica* after a duplication event and the potential of this plasticity to promote adaptation in response to low temperatures and heat stress. Overall, these results will be useful for predicting an efficient measurement to evaluate the effect of thermal stress in *G. antarctica* PI12. Further research needs to be carried out on manipulating HSP70 in *G. antarctica* PI12 as a biomarker for measuring the effects of global warming on the Antarctic Ocean.

## Figures and Tables

**Figure 1 microorganisms-09-02069-f001:**
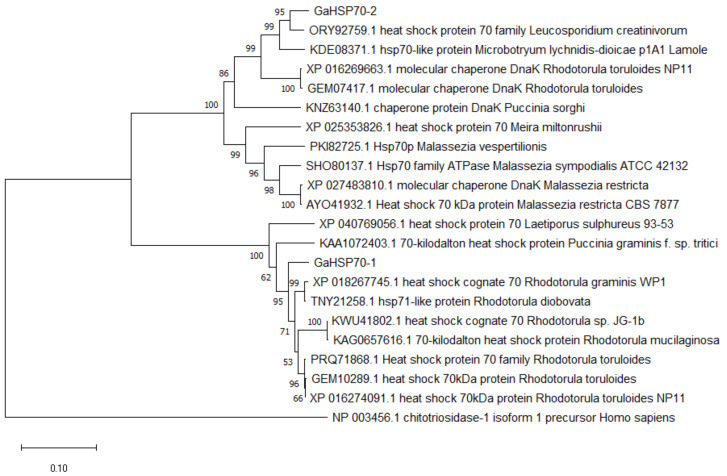
Phylogenetic tree analysis of GaHSP70-1 and GaHSP70-2 aligned with HSP70 proteins from other organisms. A chitotriosidase from *Homo sapiens* was used as an outlier. The tree was constructed using the neighbor-joining method. The numbers close to the individual branches represent the percentage of the 1000 bootstrap iterations supporting the branch.

**Figure 2 microorganisms-09-02069-f002:**
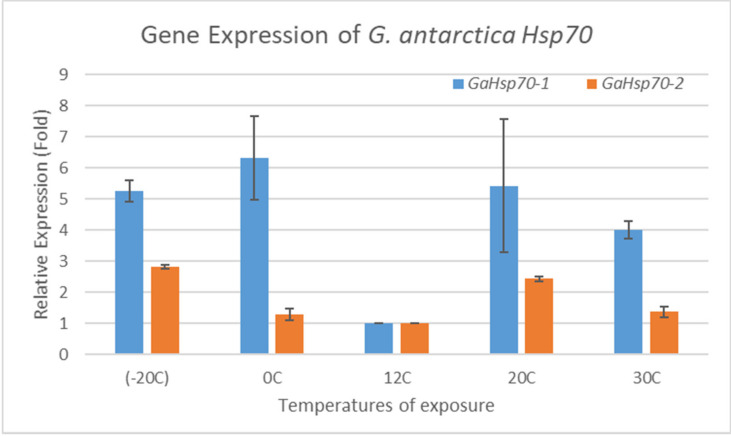
*Hsp70* mRNA expression in *G. antarctica* PI12. *GaHsp70-1* and *GaHsp70-2* levels were measured in cells exposed to temperatures of −20, 0, 20, and 30 °C for 6 h and normalized to 18 S (reference gene) levels. The mRNA expression at 12 °C was set to 1, and other values were normalized against this value. The data are representative of three trials with standard deviations of the mean, and statistical significance was assessed using a two-way ANOVA test.

**Figure 3 microorganisms-09-02069-f003:**
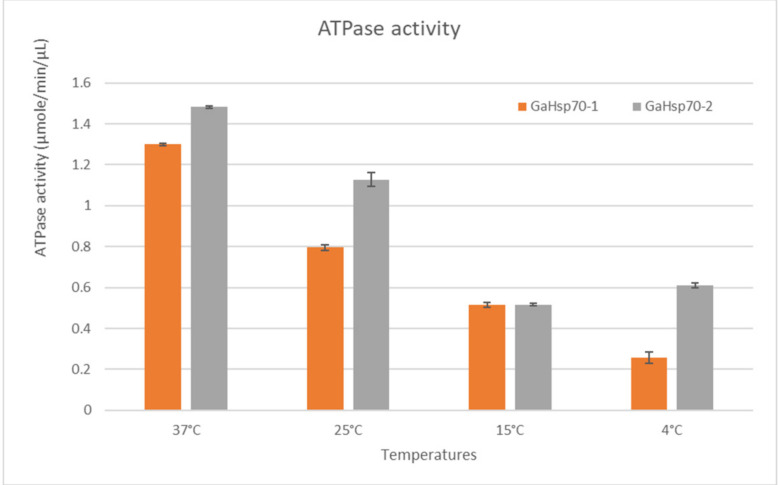
ATPase hydrolysis activity at different temperatures.

**Figure 4 microorganisms-09-02069-f004:**
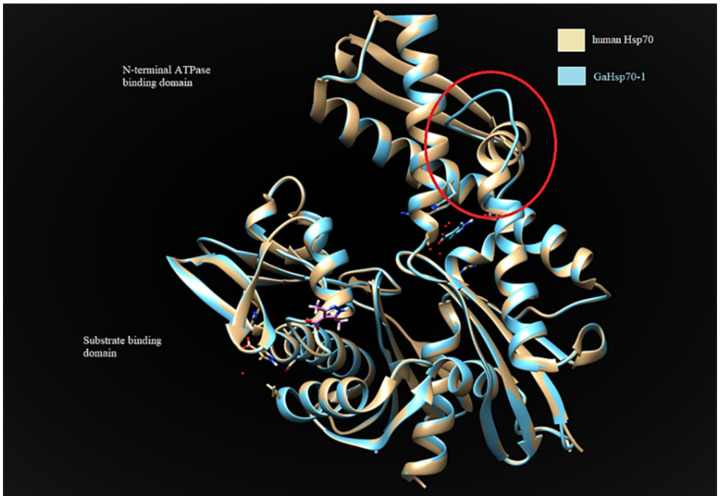
Comparative structural analysis between GaHSP70-1 model and human HSP70 (PDB 5FPN). The red circle indicates the substitution of a β-sheet for a loop located at the N-terminal ATPase of GaHSP70-1. The structures were visualized using CHIMERA UCSF [40].

**Figure 5 microorganisms-09-02069-f005:**
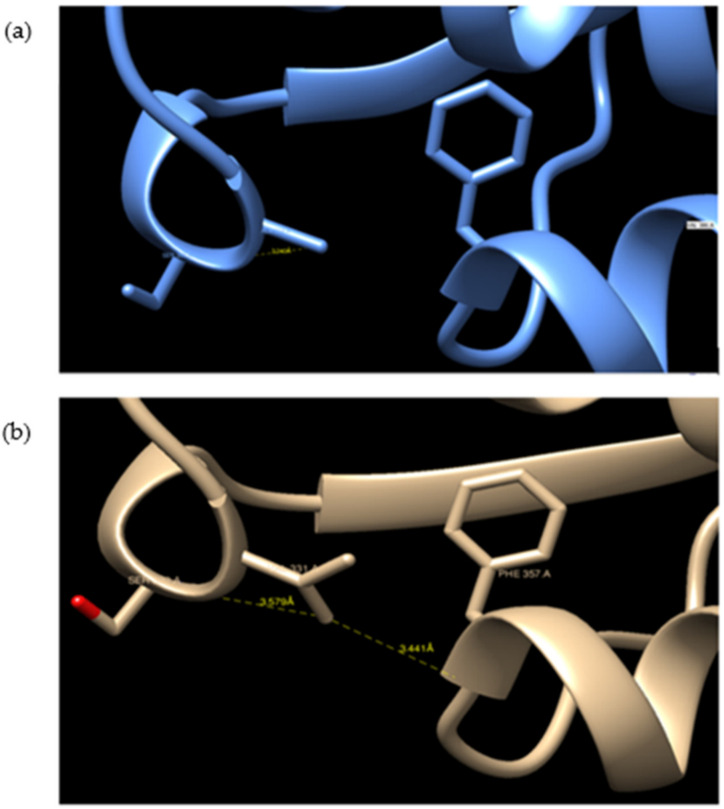
Substitution of valine for alanine in the N-terminal ATP-binding domain at position 364: (**a**) GaHSP70-2 and (**b**) DnaK. The substitution of valine for alanine hindered the aromatic interaction between valine at 364 and phenylalanine at position 357. The structures were visualized using CHIMERA UCSF [40].

**Table 1 microorganisms-09-02069-t001:** Primers used for rt-PCR.

Gene Target	Direction	Primer DNA Sequence	Size of Amplicon (bp)	R^2^	PCR Efficiency (%)
*GaHsp70-1*	Forward Reverse	5′-ATCATCGCCAACGACCAGGG-3′ 5′-CTTGGCGTCGAAGACGGTGTTG-3′	132	0.998	100%
*GaHsp70-2*	Forward Reverse	5′-AGGCTCATGTCCGCCACAAC-3′ 5′-TTGGCGGGAAGTCCAACTAATCG-3′	196	0.975	100%

**Table 2 microorganisms-09-02069-t002:** Primers used for cDNA amplification.

Gene Target	Direction	Primer DNA Sequence
*GaHsp70-1*	Forward Reverse	5′-ATGACTACCCCCGGCAAG-3′ 5′-TCA ATC GAC CTC AAC CG-3′
*GaHsp70-2*	Forward Reverse	5′-AATGCTCTCGGCAGCTCGCATTTC-3′ 5′-TTACTTCTTCTCCTCCTTGGGCTCC-3′

**Table 3 microorganisms-09-02069-t003:** Primers used for ligation independent cloning (LIC).

Gene Target	Direction	Primer DNA Sequence
*GaHsp70-1*	Forward Reverse	5′-gacgacgacaagatGACTACCCCCGGC-3′ 5′-gaggagaagcccggtCAATCGACCTCCTCA-3′
*GaHsp70-2*	Forward Reverse	5′-gacgacgacaagatAATGCTCTCGGCAGCTCGCATTTCCG-3′ 5′-gaggagaagcccggtTACTTCTTCTCCTCCTTGGGCTCC-3′

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
