# Peer review of "Characterization of Inducible HSP70 Genes in an Antarctic Yeast, Glaciozyma antarctica PI12, in Response to Thermal Stress"

_microorganisms, 2021, doi:10.3390/microorganisms9102069_

Round 1

Reviewer 1 Report

The authors of  “Characterisation of inducible HSP70 genes in an Antarctic yeast, Glaciozyma antarctica PI12 in response to thermal stress” study the expression and structure of 2 Hsp70 genes in a psychrophilic yeast. The structure and sequences of the 2 Hsp70 genes are compared in silico and a gene expression and ATPase essay was done.

Although the authors did a good job in the lab and overall the set up of experiments is conclusive the manuscript lacks structure and could be improved a lot. For example, SDS page, cloning was performed but was never mentioned in the results section while phylogenetic trees are shown in the result section and not in the material methods section. Very often basic knowledge is explained while some specific details about the performed experiments are missing. The language can be improved throughout the manuscript. Company names, references for programs are often missing. Further, it would have been a lot easier to review the manuscript if there would have been line numbers!

Please see attached pdf for further, additonal comments

Reviewer 2 Report

Yusof and coworkers present in this manuscript the characterisation of HSP70 genes from the Antarctic yeast Glaciozyma antarctica, challenged to different growth temperatures. 

The authors have a strong history on working with this psychrophilic yeast, they cited some of their own works, but omitted, maybe intentionally one paper dealing with the characterisation of G. antarctica PI12 chaperone genes expression patterns in response to heat stress (Yusof et al 2019, doi: 10.24425/ppr.2019.129674)). This published manuscript already describe 5 different HSP70 transcripts with differential localisation and expression patterns under heat shock. Whereas in the present manuscript, they only describe two, GaHSP70-1 and GAHSP70-2, without giving a gene reference number for them nor their location in the available genome.  

For this reason, my main concern about this manuscript relies with the proper identification of the HSP70 genes in G. antarctica. Despite that the authors perform a protein alignment with Saccharomyces cerevisiae and other species including Homo sapiens (please pay attention to the Figure 1 legend (page 5), gene codes should be included and Homo sapiens should be properly spelled), they present a phylogenetic relationships of both HSP70 in two different gene trees (shown in Figure 2 in page 6), without indication of proper gene codes or not indicating why the authors not used the shown alignment to produce these gene phylogenies nor why they selected those genes as the proper paralogous. This will affect the second paragraph of your discussion section (at the end of page 9).

It should be noted that many yeast had overcome a whole genome duplication process or small-scale duplications that give rise to some duplicates as can be these HSP70 onhologs or paralogs. In addition, Saccharomyces cerevisiae has two HSP70 genes, Ssa1 and Ssa2, showing functional divergence after the duplication event. So, which one was used to create the alignment and the phylogenetic tree shown in Figure 2? 

I would ask the authors to check this possibility at the first point. And clarify which genes they used, meaning, which is the code used by  Firdaus-Raih et al. 2018 (PLoS ONE, describing the genome) for each HSP70 gene, and how these genes map to the transcripts identified as 1 to 5 by Yusof et al. 2019 (Polish Polar Research), and to the sequences they labelled as GaHSP70-1 and GaHSP70-2. And how these map to the reference genome of Saccharomyces cerevisiae, and other reference sequences.

I have checked the GenBank accession ASRT0000000, and it gives no results, I found something linked to it, PRJNA202387 a Bioproject with indication of this accession number. So, it's a bit hard to know, where in the genome whole shotgun sequencing scaffold there are the HSP70 genes.  

The authors should also indicate what means the red letters in the alignment, specially in GaHSP70-2, and why they selected to use the human HSP70 as "outgroup"? 

At page 7, point 2.4. 3D structure ..., the authors indicate that GaHSP70-1 and GaHSP70-2 were 60% similar, to me, this means that if performing a good alignment PAML, they could find that both genes are paralogs, coming from a duplication event. And this will improve their findings, all the non-synonymous changes described in this result section maybe could explained by the functional divergence of copies after a duplication event, that could explain also their expression levels plasticity and use in stress response.  

In addition to these concerns, I would like to suggest the authors to refocus the introduction. In its actual state, it is really repetitive. The other introduction of the authors other papers seems more targeted and focused than this one. 

Some minor English spelling corrections should be applied throughout the text. 

Round 2

Reviewer 1 Report

The changed manuscript increased a lot in its quality and the authors did a very good job in addressing all comments.

I just have 2 more small comments to address:

-Please when referring to the supplementary material refer also to which number in the file.

-Figure 1:Please change figure caption to a more detailed description. Do you still have a/b? if so please describe and also add c). There is no need to write heat shock protein for every branch - it could be simplified. Please try to use a similar presentation form for all 3 trees: in one [] are used, in the other one not – it is hard to follow. The 0,09 length scale should not be in the tree.

Author Response

Our manuscript has been reviewed by experts in the field and both reviewers
are satisfied with our revisions, with only Reviewer2 requesting Minor
edits.

Thanks

Reviewer 2 Report

Yusof and coworkers have made a great effort to improve the manuscript, following almost all my comments, and replying to all my concerns. Great work. 

However, I still have some comments to add, that I hope the authors could address. There are two minor changes: 

line 156; Table 2 should be Table 1.

line 165; figure 3 should be Figure 2, isn't it?? as is the second figure shown up in the manuscript, as appear in page 8. 

And as final recommendation, I will suggest authors to use a proof reading service with a strong English background. I understand that authors used a Malaysian one, as they will understand what they want to tell by language proximity, but, some times the re-writing they do is not at all following the English style ... I suggest to use a good one which can be based ie. in Australia, as they should have also Asiatic translators. A cheap and fast one is Papertrue, which is located in UK, but they have also freelance translators from Asiatic countries to understand their needs.    

Author Response

Response to Reviewer 2 Comments

Point 1: line 156; Table 2 should be Table 1

Response 1: Thank you for taking the time to check our manuscript. We really appreciate it.

We have checked the whole manuscript and amended all Table list.

 Page 3, Line 117

We have changed from Table 2 to Table 1.

“….. with R2 values ranged from 0.97 to 1.00 (Table 12).”

Page 10, Lines 358-359

We have changed from Table 3 to Table 2.

Specific primers were used to amplify the cDNA targeting GaHsp70-1 and GaHsp70-2 (Table 23).

Page 10, Line 365

We have changed the from Table 3 to Table 2.

“Table 23. Primers used for cDNA amplification.”

Page 10, Line 376

We have changed from Table 3 to Table 2.

“…….in the reactions (Table 32).”

Page 11, Line 395

We have changed from Table 4 to Table 3.

“……..vector were used (Table 34).”

Page 11, Line 406

We have changed from Table 4 to Table 3.

“Table 34. Primers used for Ligation Independent Cloning (LIC).”

Point 2: line 165; figure 3 should be Figure 2, isn't it?? as is the second figure shown up in the manuscript, as appear in page 8. 

Response 2: Thank you for taking the time to check our manuscript. We really appreciate it.

We have checked the whole manuscript and all Figure list is correct except as mentioned above.

Page 3, Line 124

We have changed from Figure 3 to Figure 2.

“…..genes (Figure 32).”

Again, we thanked the reviewer for his/her efforts towards improving our manuscript.

Point 3: And as final recommendation, I will suggest authors to use a proof reading service with a strong English background. I understand that authors used a Malaysian one, as they will understand what they want to tell by language proximity, but, some times the re-writing they do is not at all following the English style ... I suggest to use a good one which can be based ie. in Australia, as they should have also Asiatic translators. A cheap and fast one is Papertrue, which is located in UK, but they have also freelance translators from Asiatic countries to understand their needs.  

Response 3: We have contacted Papertrue and received the price quotation from them. The estimated time for the accepted price is 72 hours of proofreading. We can send our manuscript for proofreading to Papertrue but we wish to get 3-5 days from Editor for amendment. We will wait for the Editor’s instruction regarding this reviewer’s suggestion.
